

**Multiple mechanisms for chlorophyll-a concentration variations in coastal**
**upwelling regions: A case study east of Hainan Island in the South China Sea**
Junyi Li [1,2,3], Min Li [1*], Chao Wang [1], Quanan Zheng [1,4], Ying Xu [3], Tianyu Zhang [1] Lingling Xie [1*]
[1] Laboratory of Coastal Ocean Variation and Disaster Prediction, Guangdong Ocean University,
Zhanjiang 524088, China
[2] Key Laboratory of Climate, Sources and Environments in Continent Shelf Sea and Deep Ocean,
Zhanjiang 524088, China
[3] Key Laboratory of Space Ocean Remote Sensing and Application, MNR, Beijing, 100081, China
[4] Department of Atmospheric and Oceanic Science, University of Maryland, College Park, MD
20742, USA
Corresponding author.
E-mail address: M. Li (min_li@gdou.edu.cn), L. Xie (xiell@gdou.edu.cn);
**Abstract**
Using satellite observations from 2003 to 2020 and cruise observations in 2019 and 2021,
this study reveals an unexpected minor role of upwelling in seasonal and interannual variations in
chlorophyll-a (Chl-a) concentrations in the coastal upwelling region east of Hainan Island (UEH)
in the northwestern South China Sea (NWSCS). The results show strong seasonal and interannual
variability in the Chl-a concentration in the core upwelling area of the UEH. Different from the
strongest upwelling in summer, the Chl-a concentration in the UEH area reaches a maximum of
1.18 mg m$^{-3}$ in autumn and winter, with a minimum value of 0.74 mg m$^{-3}$ in summer. The
summer Chl-a concentration increases to as high as 1.0 mg m$^{-3}$ with weak upwelling during El
Niño years, whereas the maximum Chl-a concentration in October increases to 2.5 mg m$^{-3}$ during
La Niña years. The analysis of environmental factors shows that compared to the limited effects of
upwelling, the along-shelf coastal current from the northern shelf and the increased precipitation
are crucially important to the Chl-a concentration variation in the study area. These results provide
new insights for predicting marine productivity in upwelling areas, i.e., multiple mechanisms,
especially horizontal advection, should be considered in addition to the upwelling process.

**Keywords**: Coastal upwelling; chlorophyll-a concentration; Guangdong Coastal Current; ENSO
events; EOF analysis

**1 Introduction**
The oceanic area with coastal upwelling is generally characterized by high productivity; it
occupies only 1% of the total area of the ocean but provides more than 50% of the total marine
fish harvest (Barua, 2005). High levels of biological productivity strongly influence atmosphere-
ocean carbon recycling (Mcgregor et al., 2007; Xu et al., 2020). Therefore, revealing the variation
in chlorophyll-a (Chl-a) in coastal upwelling areas is important to the overall health of the marine
ecosystem and climate.
The upward movement of seawater may carry nutrients from the lower layer and support a
high surface Chl-a concentration. Thus, the variability in Chl-a concentrations in coastal upwelling
regions is proposed to be associated with that of upwelling (Jing et al., 2009). Alongshore winds,
positive wind curl, tidal mixing and topography may affect upwelling processes (Hu and Wang,

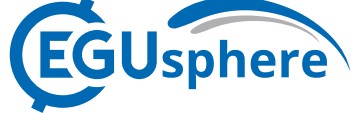

2016). In contrast, other oceanic and atmospheric processes, such as mesoscale eddies, sub-
mesoscale fronts, precipitation and typhoon processes, can also induce Chl-a increments (Aoki et
al., 2019; Cape et al., 2019; Li et al., 2021a; Li et al., 2021b).
The coastal upwelling east of Hainan Island (UEH) is part of the seasonal upwelling in the
northwestern South China Sea (NWSCS). As shown in Figure 1, the isobaths in the shelf are
parallel to the continental coastline. The width of the continental shelf is approximately 100 km.
Outside of the continental shelf, there is a steep slope linking the shelf to the South China Sea
(SCS) Basin. The circulation in the coastal area east of Hainan Island (HEC) is controlled by the
East Asian monsoon system. In summer, the coastal current travels northeastward on the shelf
influenced by the southwesterly monsoon, whereas in winter, the current flows southwestward
(Ding et al., 2018; Jing et al., 2015). According to the Ekman transport theory, the along-shelf
wind induces cross-shelf transport of the surface water and thus causes coastal upwelling along the
coastline in summer. The UEH generally begins in April, becomes strongest in July and August,
and remains until September (Xie et al., 2012). The UEH is located in coastal shallow water less
than 100 m (Jing et al., 2015). Wind stress curl-induced Ekman pumping is considered to be
another crucial factor for UEH generation (Xu et al., 2020). In addition, the strong northeastward
current along the shelf enhances upwelling (Su et al., 2013).

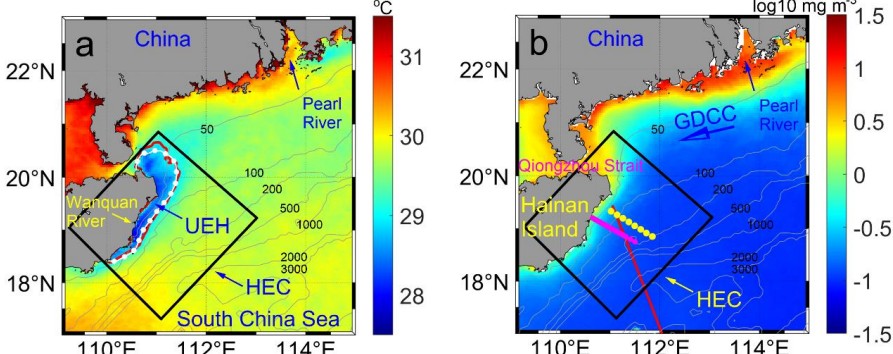


Figure 1. Study area (black solid square) and sampling sites. (a) Climatological (June-August) sea
surface temperature (SST) and (b) Chl-a concentration during 2003–2020. In panel (a), the white
dotted curve is the SST front for June-August; the red curve is the 29° isotherm. In (b), the dots
are the observation sites for the cruise during July 14–15, 2021 (yellow), and October 2–3, 2019
(magenta), and the red curve is the altimeter satellite ground track (Track 114). The unit of the
numbers on the isobaths is meters.

The variation in primary production in the HEC has been variously reported. Deng et al.
(1995) reported that phytoplankton achieved a maximum value in a strong period of UEH. Jing et
al. (2011) found a higher Chl-a concentration in summer 1998, as the offshore Ekman transport
was the strongest. Southwesterly monsoon-induced coastal upwelling is suggested to be the major
mechanism for the relatively high summertime phytoplankton biomass and primary production
(Liu et al., 2013; Song et al., 2012). Moreover, Hu et al. (2021) found that eddy processes could
strengthen phytoplankton blooms in the HEC. The variation in the basin circulation may also



affect the UEH (Su et al., 2013; Wang et al., 2006). However, Ning et al. (2004) reported poor
nutrients, low Chl-a, and weak primary production in summer in the HEC. Shi et al. (2021) found
that the largest Chl-a increase in the HEC occurs in May when upwelling is weak. Li et al. (2021a)
further showed that the maximum Chl-a concentration year-round exhibits a double peak in March
and October.

The results of previous studies indicate that upwelling may not be the most significant factor

affecting primary productivity in the HEC (Li et al., 2021a; Ning et al., 2004). The mechanism
driving the variation in primary productivity in the HEC thus needs further investigation.

The objective of this study is to reveal the role of upwelling in the spatial-temporal variations

in Chl-a concentrations in the HEC area based on multi-sensor satellite observations and in situ
cruise observations. The article is organized as follows. Section 2 describes the materials and
methods, including the algorithms used for retrieval of the total suspended sediment (TSS) and sea
surface temperature (SST) front from satellite observations. Section 3 presents the results and
analysis of the spatial-temporal variations in the Chl-a concentration in the study area. Section 4
presents variations in environmental factors in the study area, including monthly climatological
wind, rainfall, SST, sea surface salinity (SSS), euphotic depth, and TSS. Section 5 discusses the
role of typhoons, coastal currents, ENSO events, and precipitation in the Chl-a concentration.
Section 6 presents the conclusions.

**2 Materials and methods**
2.1 Study area and upwelling area

The study area (enclosed by the black square in Figure 1) covers the UEH area off the

northeastern coast of Hainan Island. It is adjacent to the narrow Qiongzhou Strait in the west and
adjoins the wide continental shelf of the NWSCS in the east. The Wanquan River flowing through
east Hainan Island is the third largest river on Hainan Island. The East Asian monsoon prevails in
the HEC, and the UEH appears along the coast in summer (Lin et al., 2016). In fall and winter, a
southwestward current flows along the coast on the whole shelf (Ding et al., 2018; Li et al., 2016).
The nutrients in the Pearl River runoff can be transported to the HEC area by the Guangdong
Coastal Current (GDCC). The thermal fronts stretch along the continental shelf (dotted white
curve in Figure 1a) and are accompanied by high Chl-a concentrations (Figure 1b).

2.2. Satellite observations and retrieval

The monthly ocean color elements (Kd490, Rrs645, Chl-a, SST, and photosynthetically

active radiation (PAR)) were obtained by moderate resolution imaging spectroradiometer
(MODIS) instruments onboard the Terra and Aqua satellites. The dataset from 2003 to 2020 is a
level-3 product with a spatial resolution of 4 km. The data from the two platforms were merged to
improve the coverage of the Chl-a concentration (Li et al., 2021b). The TSS concentration was
estimated from the Rrs645 product (Li et al., 2021b):
$$C_{\text{TSS}} = 0.6455 + 1455.7 \times Rrs645 \tag{1}$$

The euphotic depth retrieval from the Kd490 product was conducted as follows (Zhao et al.,

2013):

$$Z_{\text{eu}} = 0.28 + \frac{395.92 \times 0.0092}{0.0092 + Kd490} \tag{2}$$



The surface thermal front was estimated using the SST gradient. The SST gradient was
calculated using the zonal and meridional components (*GSSTx*, *GSSTy*) as follows:

$$GSST = \sqrt{(GSSTx)^2 + (GSSTy)^2} \tag{3}$$

where $GSSTx_i = \left(\frac{SST_{i+1} - SST_{i-1}}{x_{i+1} - x_{i-1}}\right)$ (°C/km), and $(x_{i+1} - x_{i-1})$ is equal to twice the spatial
resolution.
The sea surface wind (SSW), sea surface wind stress, and wind stress curl at 10 m above the
sea surface, with a spatial resolution of 0.25°, were obtained from the Copernicus Marine Service
(CMEMS). The wind data from May 2007 to December 2020 used in this study were a monthly
product, which was estimated from daily global wind fields calculated from retrievals derived
from advanced scatterometers (ASCAT). The wind data from January 2003 to April 2020 were
calculated from the daily global wind fields obtained by quick scatterometers (QuikSCATs).
A cross-shelf and along-shelf coordinate system for the SSW vector is given by:

$$u_{\text{along}} = u\cos\theta - v\sin\theta \tag{4}$$

$$v_{\text{cross}} = u\sin\theta + v\cos\theta \tag{5}$$

where the cross-shelf wind, $v_{\text{cross}}$, is seaward positive; the along-shelf wind, $u_{\text{along}}$, is southward
parallel to the coastline; $\theta$ is the angle between the shoreline and the north direction, 25° in this
study; and $(u, v)$ are the east and north components of the SSW.
The monthly sea surface salinity (SSS) data from 2018 to 2020 with a spatial resolution of
0.25° were obtained from the CMEMS.
The daily rainfall rate during 2003–2020 was obtained from the multi-satellite precipitation
analysis dataset of the Tropical Rainfall Measuring Mission (TRMM). The monthly data with a
spatial resolution of 0.25° were calculated from the daily global rainfall data.
The satellite altimeter along-track sea level anomaly (SLA) data from 2003 to 2020 were
obtained from the CMEMS. The Jason-1, Jason-2, and Jason-3 satellites repeat their ground tracks
every 9.9 d. Their sampling frequency is 1 Hz, and their spatial resolution is approximately 7 km.
The 5-point moving average was applied to the along-track SLA data to filter out the small-scale
ocean processes. As the coastline is almost perpendicular to ground track 114 of the altimeter
satellites (Figure 1b), the along-shelf geostrophic current was estimated from the along-track SLA
data as follows:

$$u = -\frac{g}{f}\frac{\partial\eta}{\partial y}, \tag{6}$$

where $g$ is the acceleration due to gravity, $f$ is the Coriolis parameter, and $\eta$ is the SLA.
The typhoon track data were downloaded from the Tropical Cyclone Data Center of the
China Meteorological Administration (CMA). The dataset contains 6-hourly tracks and intensity
analyses of typhoons that occurred in the western North Pacific from 2003 to 2020.

2.3. Shipboard sections
Two shipboard sections were investigated during October 2–3, 2019, and July 14–15, 2021
(red and magenta points in Figure 1b). At each station, the temperature, salinity, and fluorescence
profiles were collected using a Sea-Bird 911plus conductivity-temperature-depth (CTD) system.
The Chl-a data from the fluorescence sensor of the CTD were not calibrated, and the signals of
interest were clear.




2.4. Mapping the upwelling

The thermal fronts (Figure 1a) of the climatological SST in the summer stretched along the
29°C isotherm. Thus, we defined the upwelling domain, i.e., core upwelling, as the area where the
SST was lower than 29°C in the summer. The time series of the core upwelling area was
calculated for each year during 2003–2020. Then, the time series of the Chl-a concentration in the
core upwelling area for each year was obtained.

The upwelling index (UI) based on the wind stress is as follows:

$$M_x = -\frac{\tau_y}{f\rho},$$                         (4)

where $\rho$=1025 kg m$^{-3}$ is the water density, $f$ is the Coriolis parameter, $\tau_y$ is the along-shelf wind
stress, and $M_x$ is the cross-shelf Ekman transport.

**3 Chl-a concentration variations in the UEH**
3.1 Variabilities in upwelling

The UI derived from the wind stress and wind stress curl are shown in Figure 2. The results
reveal that the UI decreased from 2003 to 2013 and increased from 2014 to 2020, probably due to
the phase switching of the Pacific Decadal Oscillation (PDO) in 2014 (Qin et al., 2018). Overall,
the UI from wind stress exhibited a decreasing trend from 2003 to 2020. However, the wind stress
curl exhibited a weak increasing trend from 2003 to 2020.

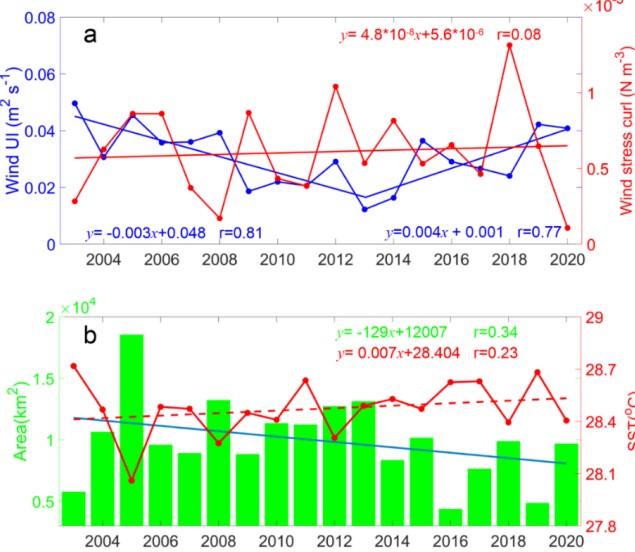


Figure 2. Upwelling index (UI) and upwelling characteristics. (a) Sea surface wind UI and wind
stress curl in the study area. (b) Upwelling area and SST. In (a), the blue dotted curve denotes the
UI of the wind stress during June-August; the red dotted curve is the time series of the mean wind
stress curl during June-August; and the blue and red curves are the trends of the UI and wind
stress curl, respectively. In (b), the green bar and red dotted curve denote the area and mean SST
for the region of the study area with temperatures of less than 29°C, respectively, and the black


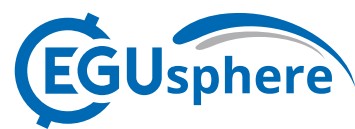

and red curves are the trends of the area and mean SST, respectively. The blue curve with squares
denotes the UI of SST during June-August. The blue curve is the trend of SST UI.
The area and SST of the UEH are shown in Figure 2b. The time series of the area of UEH
exhibited a downward trend from 2003 to 2020. Moreover, the mean SST in the UEH exhibited an
increasing trend. The trends of both the area ($-129$ km$^2$ y$^{-1}$) and mean SST ($0.007$°C y$^{-1}$) indicate
that the UEH gradually weakened from 2003 to 2020.
The time series of the UEH area and UI exhibit interannual variations. High UI values
occurred in 2005, 2008, 2012, and 2015, which coincided with the large areas of upwelling in
these years. Low UI values occurred in 2004, 2006, 2009, 2016 and 2019, which coincided with
the small areas of upwelling in these years. The area of upwelling was small in 2003 and 2019, but
peaks in the UI occurred in these years.
3.2. Variabilities in Chl-a concentration in the UEH
The time series of the spatial mean of the Chl-a concentration in the UEH is shown in Figure
3. The Chl-a concentration is unexpectedly low from April to September, i.e., the upwelling
season (as shown in Figures 3a-b). The climatological mean Chl-a concentration is the lowest in
summer, 0.74 mg m$^{-3}$ (as shown in Figure 3b and Table 1), which indicates the relatively limited
effect of upwelling on the Chl-a concentration in the HEC. However, the mean Chl-a
concentration in the UEH is highest in autumn (1.18 mg m$^{-3}$) and almost twice as high as that in
summer. In October, the mean Chl-a concentration is as high as 1.4 mg m$^{-3}$.

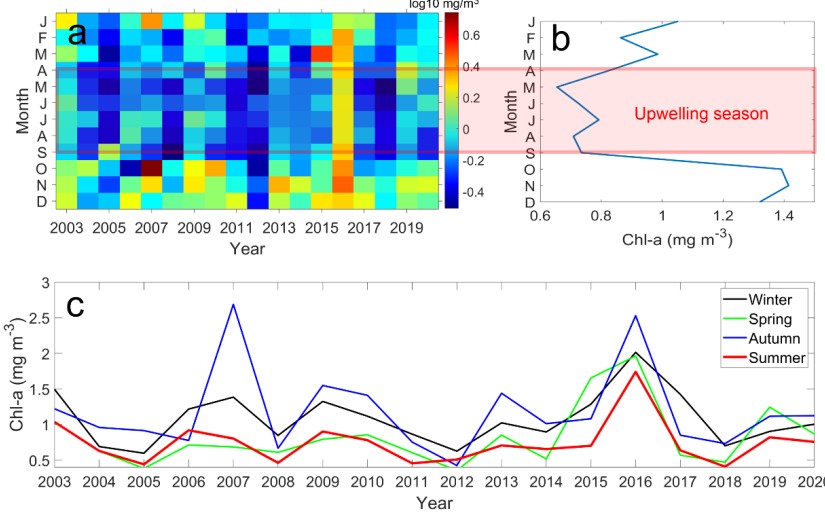

Figure 3. Time series of (a) the spatial mean of the Chl-a concentration in the upwelling area, (b)
the monthly climatological mean, and (c) the seasonal mean. The red shading indicates the
upwelling season from April to September.
Table 1. Seasonal climatologic mean Chl-a concentration in the UEH.

| Period | Winter | Spring | Summer | Autumn | Annual mean |
|--------|--------|--------|--------|--------|-------------|
| Value | $1.08 \pm 0.24$ | $0.82 \pm 0.44$ | $0.74 \pm 0.06$ | $1.18 \pm 0.23$ | $0.96 \pm 0.27$ |






The interannual variations in the spatial mean of the Chl-a concentration in the UEH are also
shown in Figure 3. The Chl-a concentration in the UEH was high in 2003, 2006–2007, 2009–
2010, 2013, 2015-2016 and 2019. The Chl-a concentrations in these years were 2–4 times
(ranging from 1.0 to 1.8 mg m$^{-3}$) those in the other years (2005, 2008, 2011-2012 and 2017-2018).
In the remaining years, the Chl-a concentration is only approximately 0.5 mg m$^{-3}$ in summer,
which is much less than the mean value.
Comparing the time series of Chl-a concentration shown in Figure 3 to the time series of
upwelling characteristics shown in Figure 2, one can see that low UI values coincide with high
Chl-a concentration in the UEH, and vice versa. It is known that low UI values indicate weak
upwelling in the HEC. This means that upwelling is a limiting factor in the UEH. Moreover, one
can see that the Chl-a concentration is unexpectedly low in the upwelling season, as shown in
Figures 2a-b. Therefore, the results provide new insight into the relationship between marine
productivity and upwelling in the UEH. However, the effect of environmental factors and spatial
variations on the Chl-a concentration need further investigation.

**4 Variations in environmental factors in the HEC**
4.1 Monthly variations
Figure 4 shows the climatological monthly variations in the environmental factors in the
study area. As shown in Figure 4a, the mean along-shelf component of the SSW is positive from
May to September, with the strongest value of 2.5 m s$^{-1}$ occurring in June. In the rest of the
months, the along-shelf components of the SSW and wind stress curl are negative. The cross-shelf
component of the SSW is also negative. The changes in the wind direction show that the study
area is mainly controlled by the Asian monsoon. The period of UEH is coherent with that of the
positive along-shelf wind and the wind stress curl from May to September (green shading in
Figures 4a–d), indicating the effects of SSW and wind stress curl on coastal upwelling.



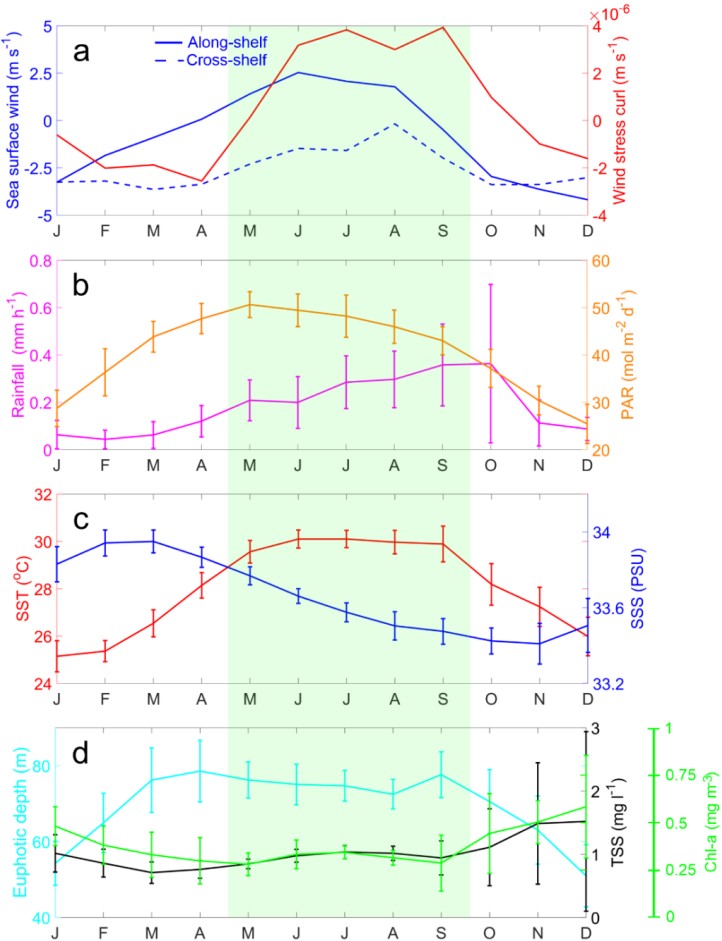

Figure 4. Monthly climatological (a) sea surface wind and wind stress curl, (b) rainfall and PAR, (c) SST and SSS, and (d) euphotic depth and TSS in the study area. The error bar indicates the standard deviation (STD). The shaded area indicates the upwelling season.

The rainfall in the study area increases monotonically from February to October and peaks in October with a value of 0.37 mm h$^{-1}$ (Figure 4b). After October, the rainfall decreases rapidly to 0.10 mm h$^{-1}$ in November. The rainfall in winter (December, January, and February) was less than 0.10 mm h$^{-1}$. Different from the rainfall, the mean photosynthetically active radiation (PAR) in the study area reaches its maximum value of 50 mol m$^{-2}$ d$^{-1}$ in May, indicating its dependence on the annual movement of the sun. The monthly climatological distribution of the SST is similar to that of the PAR, while the highest (lowest) SSS occurred in March (October and November) following the amount of rainfall (Figure 4c).

For the euphotic depth, the average values in the study area are greater than 50 m all year around and reach 70 m in the months of March to October (Figure 4d). In contrast, the TSS concentration is less than 1.0 mg l$^{-1}$ from January to September and reaches the highest value of 1.5 g l$^{-1}$ in December. Similar to the TSS, the mean Chl-a concentration in the study area has





smaller values of less than 0.3 mg m$^{-3}$ from March to September, although the UEH occurs in the
summer months (green shading in Figure 4).

4.2 Spatial distribution

Figure 5 shows the spatial distributions of seasonal climatological environmental parameters.

The PAR is almost homogeneous in the study area (Figure 5a). The values are approximately 20–
30 mol m$^{-2}$ d$^{-1}$ in winter, reach 50 mol m$^{-2}$ d$^{-1}$ in the spring and summer, and then decrease to
approximately 30–40 mol m$^{-2}$ d$^{-1}$ in autumn.

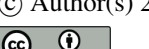

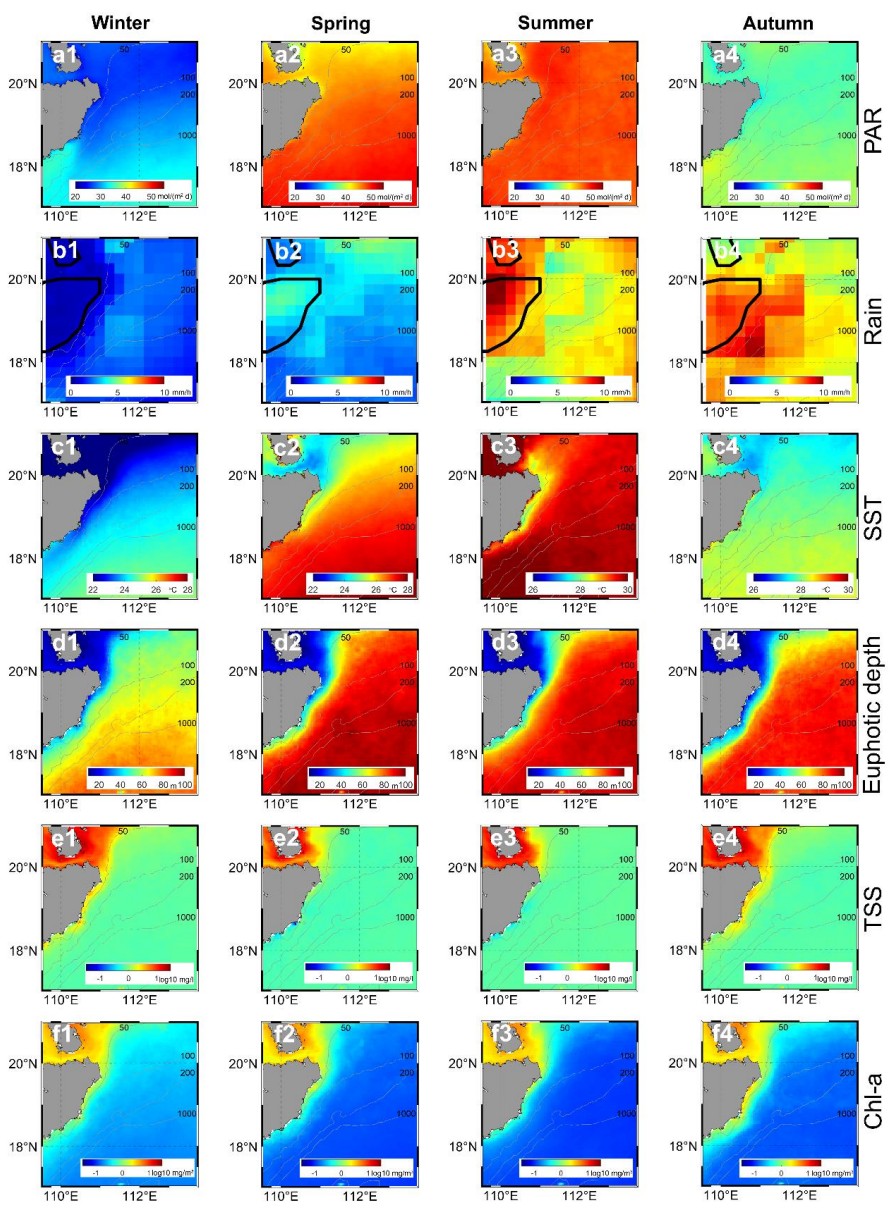

Figure 5. Seasonal climatological (a1–a4) PAR, (b1–b4) rainfall, (c1–c4) SST, (d1–d4) euphotic depth, (e1–e4) TSS, and (f1–f4) Chl-a. The columns correspond to winter, spring, summer and autumn. The unit of the numbers on the isobaths is meters.

The rainfall rate is less than 5 mm h$^{-1}$ in winter (Figure 5b). In spring and summer, the rainfall peaks in Hainan Island, while the high precipitation area is located on Hainan Island and in the HEC area in autumn. The rainfall rate is as high as 10 mm h$^{-1}$ in summer and autumn. Furthermore, as the high precipitation area is located on land, the heavy rain is transformed into



runoff, which carries nutrients into the sea. Thus, the temporal and spatial variations in the rainfall
rate likely induce variations in the input of terrestrial materials.

The SST exhibits remarkable seasonal variability (Figure 5c). Generally, the SST is high in
spring and summer and low in winter and autumn. Moreover, the SST is lower in coastal waters
than in ocean areas in winter and spring, which is modulated by the prevailing southwestward
current along the coastline of Guangdong (Ding et al., 2018). In summer, a region identified by
low SST (<29°C) values, i.e., the UEH, is observed to the northeast of Hainan Island.

The spatial distribution of the euphotic depth is consistent with the bathymetric distribution
(Figure 5d). The euphotic depth in spring and summer is approximately 20–30 m within water
depths of less than 50 m, whereas it is approximately 100 m in the deeper water. In winter and
autumn, the euphotic depth is 20–30 m within water depths of less than 70 m. Moreover, the
euphotic depth decreases to 60–80 m in the deeper water. These variations in the euphotic depth
likely affect the vertical distribution of phytoplankton in the water.

Similarly, the TSS concentration is higher in the coastal area and lower in the ocean area
(Figure 5e). Moreover, the TSS concentration is less than 0.3 mg l$^{-1}$ in spring and summer in the
HEC area. However, the TSS concentration increases to 3.0 mg l$^{-1}$ at water depths of less than 70
m in autumn and winter.

The Chl-a concentration is higher in the coastal area than in the open ocean area (Figure 5f).
In winter and autumn, the Chl-a concentration is higher than 1.0 mg m$^{-3}$ at water depths of less
than 70 m. In spring, the Chl-a concentration decreases to 0.5 mg m$^{-3}$. However, the Chl-a
concentration decreases to approximately 0.3 mg m$^{-3}$ in summer. In addition, the high
concentration is approximately 1.0 mg m$^{-3}$ in the nearshore area with water depths of less than 20
m.

4.3 EOF analysis of Chl-a concentration

To further reveal the variations in the Chl-a concentration in the HEC, the empirical
orthogonal function (EOF) analysis results are shown in Figures 6–7. The first four EOF modes of
the Chl-a concentration explain 60% of the total variance (Figure 6). Mode 1 includes an enhanced
signal in the coastal waters (<60 m) to the east of Hainan Island. The magnitude of the variability
is generally the same throughout the other areas. The corresponding temporal evolution (Figure
7a) is characterized by strong seasonal cycles, with peaks in October and troughs in May. The
climatological mean of the corresponding temporal evolution is negative from April to September
and positive from October to March. The negative phase with a large amplitude lasts for six
months. Therefore, the Chl-a concentration is persistently low from April to September. Mode 1 is
characterized by the GDCC (Ding et al., 2018). Mode 2 separates the east and northeast coastal
waters of Hainan Island. The troughs of the temporal evolution of Mode 2 occur in September and
October. The climatological mean peaks in January and December. The strong signals occur in
September and October to the east of Hainan Island, which indicates that the Chl-a concentration
is controlled by rainfall (as shown in Figure 5b). The other strong signals occur in January and
December. Moreover, they are located on the north shelf of the SCS, adjacent to the Qiongzhou
Strait to the west. Thus, the result suggests that Chl-a concentration is affected by the GDCC.





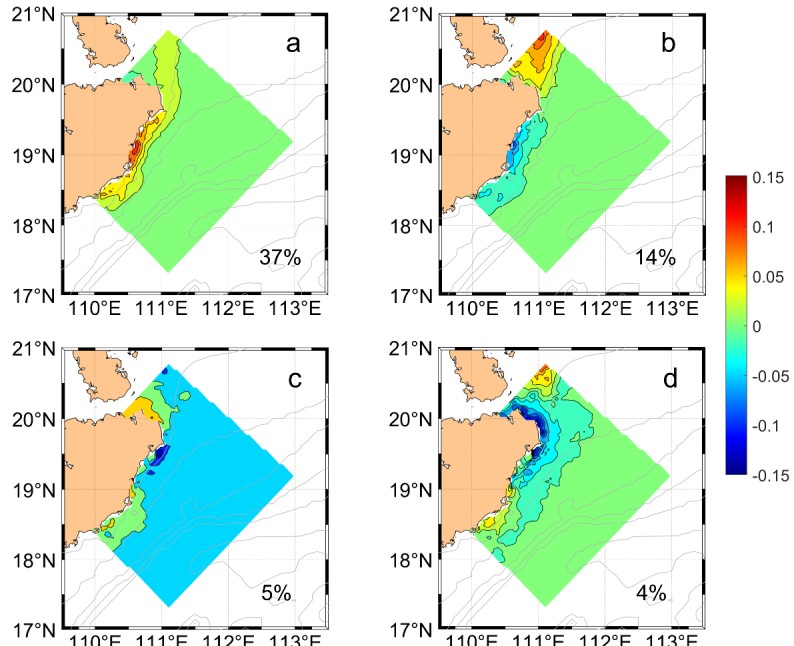

Figure 6. Spatial distributions of the first four EOFs for the Chl-a concentration. The variance explained by each mode is labeled.

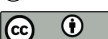


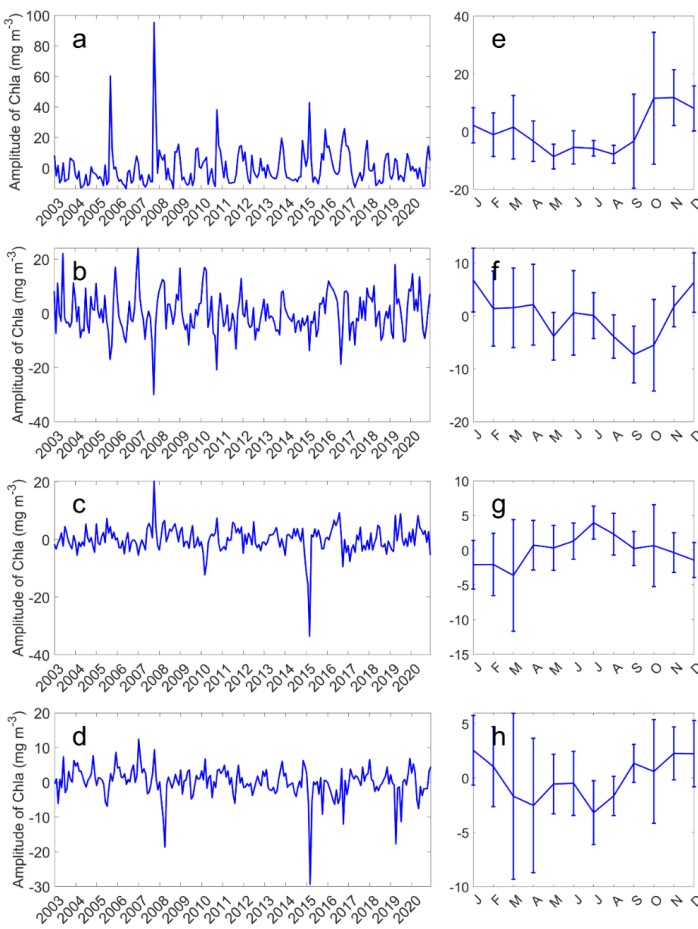

Figure 7. (a–d) Time series and (e–h) climatological mean of the first four EOFs for the Chl-a concentration.

Mode 3 describes 5% of the total variance in the Chl-a concentration in the coastal regions of Hainan Island. Mode 3 also separates the east and northeast coastal waters of Hainan Island (Figure 6c). However, the climatological mean of the temporal evolution is positive between June and August. Therefore, the positive phase occurs in summer, revealing an upwelling area to the east and north of Hainan Island. Mode 4 contributes only 4% of the total variance. The climatological mean of the temporal evolution exhibits strong peaks in July and weak peaks in April, i.e., semiannual variability. High Chl-a concentrations occur in the northeast coastal water of Hainan Island during the upwelling season.

Modes 3 and 4 both describe the upwelling phenomenon along the northeast coast of Hainan Island during summer. The spread of upwelling can be seen clearly in the EOFs of the Chl-a concentration in the areas with water depths of less than 100 m along the coastline. However, upwelling described less than 10% of the total variance in the Chl-a concentration, indicating that the contribution of upwelling to productivity in the HEC is limited.




4.4 Vertical distribution of the Chl-a concentration based on observation data

To examine the vertical distribution of the Chl-a concentration in the HEC, two cruise
measurement sections are used in this study. Figure 8 shows the oceanographic cruise data
collected on July 14–15, 2021, and October 2–3, 2019, illustrating the distribution of the Chl-a
concentration in summer and autumn, respectively. The pronounced upwelling can be seen on the
cross-shelf section observed in July 2021. Both the isotherm and isohaline on the shelf are uplifted
toward the shore by upwelling-induced movement. A temperature front can be seen near the sea
surface, which is located approximately 50 km away from the coastline (depths of ~90 m). The
thermal fronts are reported by Jing et al. (2016). The fronts induced by upwelling tend to be
approximately aligned with the 20–100 m isobath. The high Chl-a concentration layer is also
uplifted from 80 m to 40 m by upwelling, and the Chl-a concentration is as high as 1.2 mg m$^{-3}$.

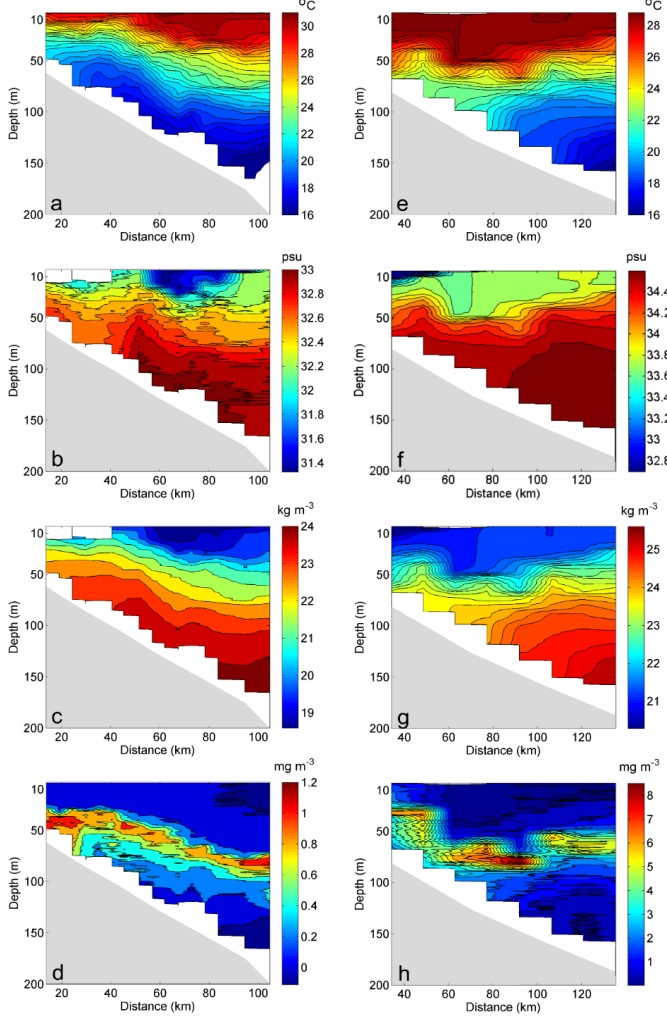




Figure 8. Oceanographic cruise data collected on (a–d) July 14–15, 2021, and (e–h) October 2–3, 2019: a) and e) temperature distributions; b) and f) salinity distributions; c) and g) potential density distributions; and d) and h) Chl-a distributions.

From October 2–3, 2019, the sea surface temperature front disappeared. Jing et al. (2016) found that the front was the weakest in autumn. However, a salinity front occurs approximately 60 km from the coastline at a depth of ~100 m. This salinity front indicates that fresh water is injected into the sea surface. Figures 4b and 5b4 show that the rainfall is strong during autumn. The rainfall is input into the sea surface via rainfall and runoff. Thus, the salinity front is generated. In contrast to the upwelling in summer, downwelling occurs in the bottom water and is associated with downwelling-favorable wind forcing. Moreover, abundant Chl-a is detected at a depth of 30 m on the shelf, which is shallower than the detection depth in summer since the euphotic depth is shallower in autumn, as shown in Figure 5d.

**5 Factors related to variations in the Chl-a concentration**

5.1 Relationship with typhoon events

In the NWSCS, the Chl-a concentration can be affected by different factors, e.g., typhoons. Typhoon-induced upwelling occasionally occurs in the SCS (Ma et al., 2021; Wang et al., 2020). In the shelf areas, typhoon-enhanced vertical mixing and upwelling play dominant roles in the spatiotemporal behavior of the Chl-a concentration (Li et al., 2021a; Li et al., 2021b). The upwelling transports nutrients into the euphotic zone, which supports Chl-a blooms (Ye et al., 2013; Zheng et al., 2021). An increase in the Chl-a concentration in the nearshore region off Hainan Island followed typhoon rainfall, with mixing and upwelling effects (Zheng and Tang, 2007). The large-scale peripheral wind vector resulted in the accumulation and enhancement of the Chl-a concentration in the nearshore area (Liu et al., 2020). An offshore bloom produced a Chl-a peak (4 mg m$^{-3}$) after the typhoon's passage (Zheng and Tang, 2007). These observations illustrate the effects of typhoons on the marine ecosystem in the HEC.

Figure 9 shows the time series of the number of typhoons that passed over the HEC during 2003–2020. Sixty-eight typhoons passed across the continental shelf of the NWSCS during this 18-year period. There were interannual variations in the time series of the number of typhoons. As many as nine typhoons were generated and affected Qiongdong in 2013, while fewer than two typhoons passed by the study area in 2004, 2007, 2010, and 2014–2015. Seasonally, 33 typhoons passed by in summer and autumn. As shown in Figure 3b, a small peak in the mean Chl-a concentration occurred in July. Moreover, the Chl-a concentration was high in 2013, especially in autumn, varied within the range of 0.7–1.5 mg m$^{-3}$, and coincided with the occurrences of nine typhoons. This indicates that the high Chl-a concentrations were related to the typhoons. However, typhoons occur on the synoptic scale and influence the coastal area for several days. Therefore, these processes seem to have a limited effect on the monthly mean Chl-a concentration.



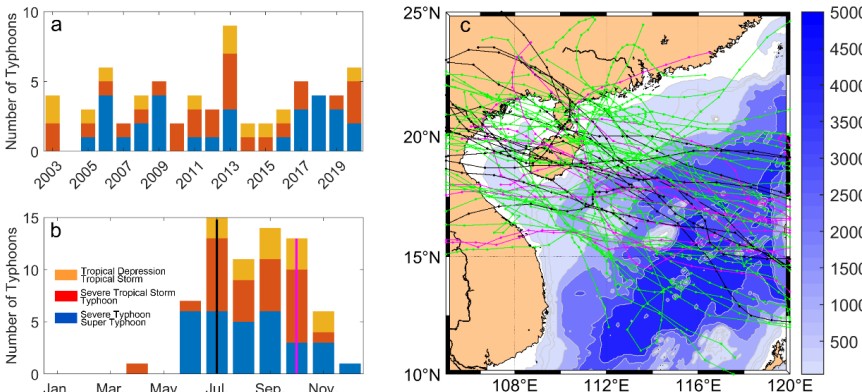

Figure 9. (a) Time series of the number of typhoons that passed by the study area during 2003–2020. (b) Seasonal distribution of typhoons. (c) Trajectories of typhoons during 2003–2020. The orange, red, and blue bars in (a–b) represent the numbers of tropical depressions and tropical storms, severe tropical storms and typhoons, and severe typhoons and super typhoons, respectively. The magenta and black curves in (c) represent the typhoons that passed by the study area in July and October, respectively. The green curves represent the typhoons that passed by the study area in the other months.

5.2 Role of the coastal current

The current in the NWSCS contributes significantly to the transport of low-salinity water, nutrients, and phytoplankton, and it also affects the ecological environment (Ding et al., 2018; Meng et al., 2017). The shelf circulation pattern is dominated by monsoons, tides, buoyancy forcing, and topography. Due to the changes in the wind direction, the current direction changes in the different seasons. In autumn and winter, the current in the NWSCS is predominantly southwestward. It changes northeastward in summer (Ding et al., 2018). The monsoon plays an important role in the current, which induces onshore and offshore Ekman transport on the shelf during the winter and summer monsoons, respectively. Gan et al. (2013) found that transport was induced by amplified geostrophic transport during downwelling events. Here, we used geostrophic current retrieval from along-track satellite altimeter data on the shelf of the NWSCS to reflect the role of the coastal current on the Chl-a concentration.

The latitudinal distribution of the climatological along-track SLA is shown in Figure 10. The climatological sea surface sloped considerably toward the ocean in October, November, and December. The sea surface on the shelf was lower than that in the ocean. The geostrophic current shows that the current was positive (northeastward) between April and August, and it was negative (southwestward) between October and March. The climatological geostrophic current in September was approximately 0 m s$^{-1}$, which indicates that the current direction changed frequently. The climatological geostrophic current was stronger than 0.1 m s$^{-1}$ in summer, and it was strongest in October, at approximately 0.17 m s$^{-1}$.

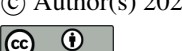



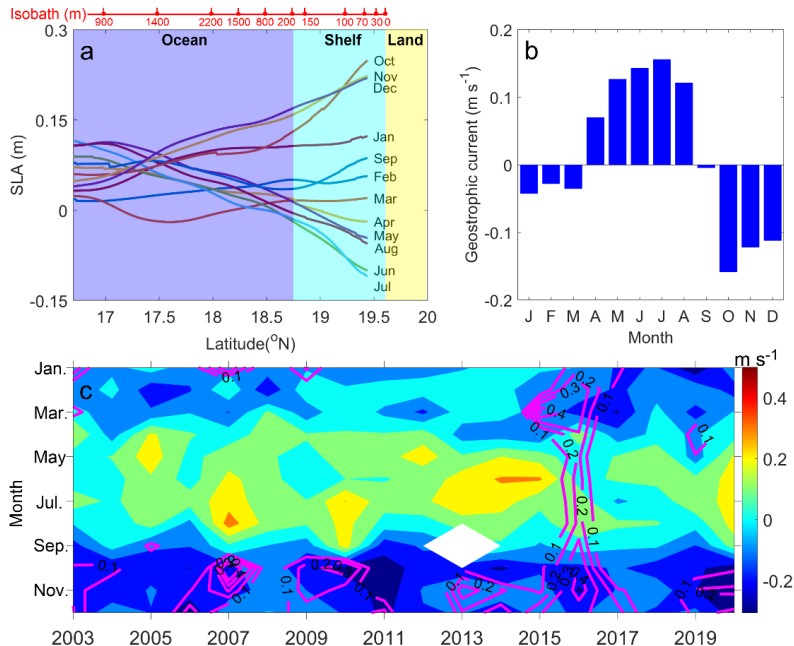

Figure 10. (a) Latitudinal distribution of climatological along-track SLA (track number: 114). (b) Geostrophic current retrieval from climatological along-track SLA. (c) Time series of geostrophic current (contours) and Chl-a concentration (magenta contour curves). The blue, cyan, and yellow shading in (a) represent the ocean, continental shelf, and land areas, respectively. The red bar with numbers in (a) indicates the water depth of the along-track SLA data. The values in (c) are the exponents of the Chl-a concentration.

In autumn and winter, the abundant nutrients in the GDCC, which were provided by the Pearl River, likely supported the high food availability to the phytoplankton (Yang and Ye, 2022). The GDCC was characterized by a high TSS (Figure 4d and 5e). TSS is synergistic with the concentration of dissolved nitrogen and is the dominant factor affecting the Chl-a concentration (López Abbate et al., 2017). The distribution of the monthly climatological Chl-a concentration (Figure 3b) was similar to that of the geostrophic current. In summer, the Chl-a concentration was low during the northeast oligotrophic current. In winter, the Chl-a concentration was high during the southwest nutrient-rich current. Figure 10c presents the time series of the geostrophic current and Chl-a concentration. The negative current and high Chl-a concentrations mainly occurred in autumn and winter, which demonstrates the crucial role of the current.

5.3 Role of rainfall

The phytoplankton responded more positively to the increased precipitation in the coastal waters (Thompson et al., 2015). Kim et al. (2014) reported that the increase in wind speed accompanied by rainfall was a major contributor to the Chl-a concentration. The precipitation directly deposits the nutrients in the air into the seawater. In addition, most of the rainfall on land runs over the land surface into the rivers and eventually into the ocean, transporting nutrients to



the ocean. Therefore, rain plays an important role in the variability of phytoplankton in coastal
waters.
Figure 11 shows the time series of the monthly mean rainfall rate and Chl-a concentration.
The rainfall rate was high in the summer and autumn, ranging from 0.3 to 1.4 mm h$^{-1}$. In October
in 2007–2017, the monthly mean rainfall rate (>0.5 mm h$^{-1}$) coincided with the high Chl-a
concentration.




Figure 11. Time series of the rainfall rate (contours) and Chl-a concentration (blue curves with text
labels). The values on the contours are the exponents of the Chl-a concentration.

Runoff is the main source of silicate in coastal waters (Zhang et al., 2003). Chen et al. (2016)
observed that the concentration of silicate was as high as 2–12 µmol l$^{-1}$ in the coastal waters of the
HEC and had a positive correlation with the Chl-a concentration. Wang et al. (2018) found that
diatoms contributed 88.11% and 85.81% of the total phytoplankton abundance in the northern
SCS in May and October, respectively. The Chl-a concentration can increase by 0.3 mg·m$^{-3}$ after a
rainfall event (Zeng et al., 2022). Moreover, in Mode 2 of the EOFs (Figures 6–7), the positive
phase of the Chl-a concentration occurred off the east coast of Hainan Island in October, which is
near the estuary of the Wanquan River. Therefore, the runoff caused by the high rainfall rate
triggered the high Chl-a concentrations in the HEC.

5.4 Relationship with ENSO events
ENSO has an indirect positive effect on the Chl-a concentration through its influences on
precipitation, winds, SST, and turbidity (López Abbate et al., 2017). In the SCS, easterly wind
anomalies and SST warming occurred in the summer following El Niño events (Yang et al., 2015).
During El Niño events, the weakened southwesterly monsoon suppresses ocean upwelling (Jing et
al., 2011; Kuo et al., 2008). The reverse occurs during La Niña events. Jing et al. (2011) found that
the significantly strengthened wind stress of the 1998 summer induced strong upwelling, and the
Chl-a concentration was much higher than in any of the other years.
In the upwelling season, i.e., summer, the wind stress was smaller during El Niño events
(Table 2) than during La Niña events (Figure 2a). The Chl-a concentration in July during El Niño
events increased to as high as 1.0 mg m$^{-3}$ (Figure 3a). Moreover, the SST of the core upwelling
area was higher, but the core area was smaller (Figure 2b). Therefore, ENSO events regulated the
Chl-a concentration of the upwelling through wind stress.

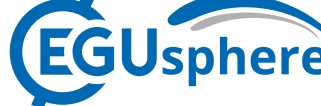

Table 2. ENSO events during 2003–2020.

| ENSO event | Years |
|---|---|
| El Niño | 2003, 2004, 2006, 2009, 2015, 2019 |
| La Niña | 2005, 2007–2008, 2010–2012, 2016, 2017–2018 |


In autumn, especially October, the spatial mean Chl-a concentration in the upwelling area
was as high as $1.18 \pm 0.23$ mg m$^{-3}$. The precipitation was heavier during La Niña events (i.e., in
2005, 2007, 2010–2011, and 2016) than during El Niño events. Furthermore, the along-shelf
current from the north was crucially important to the Chl-a concentration. There was a positive
relationship between the Chl-a anomalies and the La Niña events.

5.5 Mechanisms of Chl-a variations in the HEC area
Figure 12 shows the relationships between the geostrophic current and rainfall and the Chl-a
concentration during 2003–2020. The Chl-a concentration increased with increasing rainfall in
August, i.e., the upwelling season. The rainfall was converted to runoff and flowed into the coastal
waters. The Chl-a concentration in summer was mainly regulated by upwelling processes (Jing et
al., 2011), with a negative correlation (Figures 2–3). Therefore, the increased precipitation and
weaker upwelling processes could have induced the increased Chl-a concentration in the HEC
(upward arrow in Figure 12).

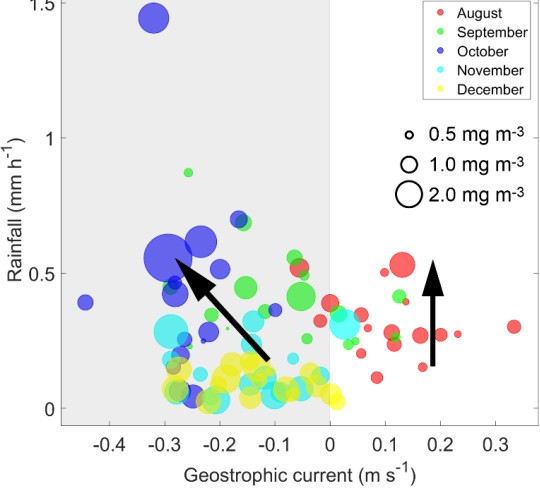


Figure 12. Bubble diagram showing the relationships between the geostrophic current and rainfall
and the Chl-a concentration. The size of the bubble represents the Chl-a concentration. The left
panel in grey represents the southwest along-shelf current in winter. The right panel represents the
northeast along-shelf current in summer. Black arrows represent the relationship between the
geostrophic current and rainfall and the Chl-a concentration.

In autumn and winter, the Chl-a concentration was increased by the increases in rainfall and



the northeastward coastal current (oblique upward arrow in Figure 12). In October, the heaviest
rainfall and the strongest current coincided with the highest Chl-a concentration. The coastward
wind component was strongest in October (Figure 4a), as was the northeast monsoon, which
induced coastward Ekman transport (Xuan et al., 2021). The downwelling movement transported
the nutrients from the rivers and the coastal current to the middle and under layers on the shelf,
which promoted an increase in silicate-favoring phytoplankton. The cruise data provide evidence
of the high Chl-a concentrations over the shelf (Figure 8h).

**6 Conclusions**
In this study, in situ observations and monthly satellite observations from 2003 to 2020 are
used to investigate the spatiotemporal variability in the Chl-a concentration in the HEC area.
Along-track satellite altimeter data for the continental shelf of the NWSCS were used to retrieve
the geostrophic current. In addition, cruise data obtained in October 2019 and July 2021 were used
to examine the vertical structure of the Chl-a concentration during the three observational seasons.
Driven by the prevailing monsoon, the SST of the core upwelling area (within a depth of 100
m) in summer increased, but its area decreased, which indicates that the UEH weakened during
the 18-year study period. The EOF analysis of the Chl-a concentration revealed that it exhibited
strong seasonal and interannual variability in the NWSCS. The climatological average Chl-a
concentration mostly peaked near the coast in autumn, 1.18 mg m$^{-3}$. However, the Chl-a
concentration in the core upwelling area was lowest during the upwelling season, approximately
0.74 mg m$^{-3}$ in summer, which contradicts the previous conclusion of a high-productivity
upwelling system.
ENSO events regulated the Chl-a concentration of the upwelling area through wind stress.
The interannual variations in the spatial mean of the Chl-a concentration were consistent with the
ENSO events. There was a positive correlation between the Chl-a anomalies and the La Niña
events. In El Niño years, the Chl-a concentration decreased to a lower level in summer. However,
the summer Chl-a concentration increases to as high as 1.0 mg m$^{-3}$ with weak upwelling during El
Niño years.
Both the along-shelf current from the north and precipitation were crucial factors controlling
the Chl-a concentration in the UEH area. The downwelling movement transported nutrients from
the rivers and the coastal current to the middle and lower layers on the shelf, which promoted an
increase in silicate-favoring phytoplankton. These results provide scientific evidence for the
development of the marine economy in the upwelling area.

**Acknowledgments**
The authors are grateful to the anonymous reviewers for their valuable suggestions and comments.
This research was funded by the National Natural Science Foundation of China (41476009,
41506018, 41976200, 41706025); Innovation Team Plan for Universities in Guangdong Province
(2019KCXTF021); and First-class Discipline Plan of Guangdong Province (080503032101,
548    231420003).

**Data Availability Statement**
Kd490, Rrs645, Chl-a, SST, and PAR data were downloaded from Ocean Color Data Processing
System (http://oceandata.sci.gsfc.nasa.gov/)





SSW, SSS, rainfall rate, and along-track SLA data were downloaded from CMEMS
(https://marine.copernicus.eu/)
The shipboard sections data are archived at https://dx.doi.org/10.6084/m9.figshare.19679538.
The typhoon track was obtained from the Tropical Cyclone Data Center of the China
Meteorological Administration (CMA) (http://tcdata.typhoon.org.cn).
**Author contributions**
JYL, ML and LLX were responsible for writing the original draft. Review and editing were
conducted by QAZ. Conceptualization was handled by JYL, QAZ and LLX. CW, YX and TYZ
were responsible for data curation. LLX acquired funding.
**Competing interests**
The contact author has declared that none of the authors has any competing interests.
**Disclaimer**
Publisher's note: Copernicus Publications remains neutral with regard to jurisdictional claims in
published maps and institutional affiliations.

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





Science and Engineering 9, 324.

