# Peer review of "Multiple mechanisms for chlorophyll-a concentration variations in coastal upwelling regions: A case study east of Hainan Island in the South China Sea"

_EGUsphere, 2022_

## Referee Comment (RC2)

**Review of the manuscript: Multiple mechanisms for chlorophyll-a concentration variations in coastal upwelling regions: A case study east of Hainan Island in the South China Sea**

This works analyzes the seasonal cycle of the chlorophyll concentration and the factors affecting to such concentrations in an upwelling region close to the Hainan Island in the NW South China Sea.

This work is interesting and well written. It is easy to read and follow. The authors use a large number of variables, including satellite chlorophyll data, wind and curl of the wind, altimetry data, rainfall rates and *in situ* temperature, salinity and chlorophyll data from two oceanographic campaigns in July 2021 and October 2019. Beside this, euphotic layer depth and total suspended sediments are estimated from satellite data.

Considering the large amount of information provided in this work, I consider that it is a very exhaustive study and it merits publication. Nevertheless, there are several questions that need clarification previous to its final acceptance.

**General comments.**

I think that the organization of the manuscript could be improved. There is a section 1. Introduction, and then a section 2, Materials and methods, with several sub-sections for each data set. Then the section 3 is Chl-a concentration variations in the UEH, section 4 is Variations in environmental factors in the HEC, and section 5 is Factors related to variations in the Chl-a concentration, with a final section 6 for conclusions.

In my opinion, section 3 should be Results. The present sections 3, 4 and 5 should be sub-sections within the Results section.

In the present manuscript, sections 3, 4 and 5 (that should be sub-sections) include results and some discussion. The extension of the discussion in the present manuscript is too large for a section devoted to results, but is too short for a section devoted to discussion. I think that the results section should deal just with results, and include a new section for discussion and conclusions.

**Detailed comments.**

Introduction.

Lines 60-61. I need some clarification for this sentence: "In addition, the strong northeastward current along the shelf enhances upwelling (Su et al., 2013)." I do not understand how the current enhances upwelling. If there is a wind blowing from the SW (towards NE), then the circulation induced by the wind will be made of an along wind current in geostrophic adjustment, and an ageostrophic circulation which is responsible for the Ekman transport. The geostrophic current can advect the water and materials upwelled by the Ekman transport, but it is not responsible for the upwelling and therefore it does not enhance upwelling. Isn't it? Anyway, I find the introduction well written and make it clear the objectives of the work.

Material and methods.

Lines 106-107. "…and are accompanied by high chlorophyll-a concentrations". I would say "relative high concentrations". These concentrations are high if compared with the open sea, so they are high, relative to the open sea. But the highest chlorophyll concentrations in figure 1b are observed in the river Pearl and to the SW of river Pearl, in the continental coast.

Lines 127-130:

"The wind data from May 2007 to December 2020 used in this study were a monthly product, which was estimated from daily global wind fields calculated from retrievals derived from advanced scatterometers (ASCAT). The wind data from January 2003 to April 2020 were calculated from the daily global wind fields obtained by quick scatterometers (QuikSCATs)."

I suppose that the data from QuickSCATs are for the period January 2003 to April 2007, instead of 2020. Otherwise the two dataset overlap.

Lines 159-160: "…The Chl-a data from the fluorescence sensor of the CTD were not calibrated, and the signals of interest were clear." What is the meaning of this sentence? Clear?

Lines 168-171.

In equations (4) and (5), the Cartesian components of the wind u, v (eastwards, northewards) are transformed to u-along and u-across, using the coastline as x-axis. Now you use the equation: $Mx = -\tau y/f\rho$ for the Ekman transport normal to the coast. I think that you should use cross-transport and along-coast wind stress, to be consistent with the previous definition.

Lines 289-290. "These variations in the euphotic depth likely affect the vertical distribution of phytoplankton in the water." Obviously light availability is one of the main factors controlling phytoplankton growth, but the inverse is also true. The phytoplankton abundance affects the light penetration as phytoplankton absorbs light.

Figure 5. Figures f1 to f4 show the chlorophyll concentrations for the four seasons of the year. But the four figures look exactly the same. I guess it could be because of the use of logarithmic scale? Please re-make this figure to enhance the differences between different seasons, or explain why all the seasons have the same chlorophyll distribution.

Line 302. "4.3 EOF analysis of Chl-a concentration".

The authors present results from EOF analysis in this section, but it was not explained in the Materials and methods. Obviously it is a very well known statistical technique, but a brief description should be included. For instance, if you have "n" satellite chlorophyll images with "m" pixels each image, one possibility is to consider the "m" time series of chlorophyll corresponding to the "m" pixels. Each time series has "n" data. In order to obtain the EOFs, you calculate the m x m covariance matrix. Is this the procedure you followed? Previous to the calculation of the covariance matrix you can subtract a mean value, or the seasonal cycle? How did you do it? I guess you did not subtract the seasonal cycle because your first EOF accounts for it, but these things should be explained.

Line 359 and followings. "a salinity front occurs approximately 60 km from the coastline at a depth of ~100 m. This salinity front indicates that fresh water is injected into the sea surface". I do not see this clearly. I see an area of fresh water ~32.8 from the coast to 60 km at the sea surface. Therefore this front would be located at the surface, not at 100 m. At 60 km and at 50 m depth, isohalines are almost vertical and this constitutes a salinity front. But this front is different from the front at the sea surface. The latter is caused by the upward displacements of isohalines, not but the injection of fresh water. In my opinion, the origin of this frontal area at 50 m depth seems to have a dynamical origin. At 100 km, the isohalines are in a vertical position again. Hence, it looks as if there is a depression or sinking of isohalines and isopycnals between 50 and 100 km, suggesting the presence of an anticyclonic circulation cell.

Role of coastal current.

I need some explanations in this section because I do not understand it very well. In lines 415-417, the authors say: "The climatological sea surface sloped considerably toward the ocean in October, November, and December. The sea surface on the shelf was lower than that in the ocean". But in figure 10a, I see just the opposite. The curves for October, November and December are higher at the shelf than in the ocean. Notice that the orientation of this figure is different from figure 8. In figure 8 land is to the left and in figure 10 land is to the right. According to my interpretation and equation (6):

$u = -g/f \; \partial\eta/\partial y$   u is positive in October, as you state in line 422 (0.17 m/s). But if the sea surface was lower at the shelf, then the $\partial\eta/\partial y$ would be positive, as y-axis is positive seaward (Materials and methods), and the u component of geostrophic velocity would be negative. Furthermore, the distribution of isopycnals in figure 8g suggests that the sea surface is higher at the shelf. Isopycnals of 24 kgm-3 and higher sink towards the coast. Therefore lighter water sinks at the coast and sea level must be higher to compensate for it, if a no-motion reference level exists below (basic assumption of geostrophy).

Maybe the problem comes from the following sentence: "The geostrophic current shows that the current was positive (northeastward) between April and August". According to lines 134-135: "where the cross-shelf wind, $v$cross, is seaward positive; the along-shelf

wind, $u$along, is southward parallel to the coastline". I understand that southward is positive. In April to August, the sea level increases seawards according to figure 10a. Then $\partial\eta/\partial y$ is positive and according to (6), u is negative, that means that the current is directed northwards, in agreement with the sign criterion of materials and methods and coinciding with the summer monsson.

Maybe I have not understood this properly, but I need some clarification.

---

## Author Comment (AC1)

Dear Editor and Reviewers

We are very pleased to have your comments concerning our manuscript entitled "Multiple mechanisms for chlorophyll-a concentration variations in coastal upwelling regions: A case study east of Hainan Island in the South China Sea" (egusphere-2022-969). Thank the editor and reviewers for taking time out of your busy schedule to review our paper and provide constructive comments on it. Those comments are all valuable and helpful. We have read and deal with all comments carefully. We have uploaded the file of revised manuscript with all comments highlighted with yellow shading in the text, and point-to-point responses to the reviewers' comments are present following.

**Response to Comments of Reviewer 1**

**[Major Comment 1]** The methods of EOF analysis and trend estimation were not presented in Section "Materials and methods".

**Response:** Thanks for your comment. Empirical orthogonal function (EOF) is widely used in climate research to identify dominant patterns of variability and to reduce the dimensionality of climate data. In my opinion, it is a general method. Therefore, the description is omitted in the previous manuscript. Based on your comment, a brief description has been added as Section 2.5.

2.5. Empirical orthogonal function

Empirical orthogonal function (EOF) is a useful tool and widely applied to reduce the dimensionality of climate data (North et al., 1982). EOF analysis is used to determine the dominant patterns of Chl-a in the study area. The Chl-a data is prepared as an anomaly in the form of matrix, X. Decomposition is applied by B·E = X. EOF modes (i.e., E, spatial patterns) and their corresponding principal components (i.e., B, temporal coefficients) could be obtained by decomposition of the anomaly matrix. The EOF patterns and the principal components are independent.

Reference

North, G.R., Bell, T.L., Cahalan, R.F., Moeng, F.J., 1982. Sampling Errors in the Estimation of Empirical Orthogonal Functions. Monthly Weather Review 110, 699-706.

**[Major Comment 2]** Why did the authors use two wind datasets? The results related to the surface wind are merged from the two datasets? If so, it should be mentioned in the text.

**Response:** Thanks very much for this useful comment. The data used in this manuscript is mainly remote sensing data. Sea surface wind data observed by satellite is the first choice. However, the lifetime of satellite is limited, about several years. Therefore, I used the monthly product obtained from multiple scatterometers, i.e., ASCAT and QuikSCAT. The rationality of analysis from the combined data should be discussed. However, it is not the purpose of this manuscript. Therefore, I used sea surface wind data from ERA5 with your comment. The description of the data and correlated analysis are corrected.

**Data description**

The sea surface wind (SSW) at 10 m above the sea surface, with a spatial resolution of 0.25°, were obtained from the Copernicus Marine Service (CMEMS). The wind data is a sub set from the

fifth generation European Centre for Medium-Range Weather Forecasts (ECMWF) atmospheric reanalysis of the global climate covering the period from January 1950 to present. The data from 2002 to 2020 used in this study were a monthly product

The wind stress is determined as

$$\tau = \rho_a C_D U |U|$$

where $\rho_a$, $C_D$, and $U$ are air density, drag coefficient and sea surface wind. $\rho_a = 1.29$ kg m$^{-3}$. $C_D = (0.75+0.067U) \times 10^{-3}$ (Garratt, 1977). Moreover, wind stress curl is obtained by $\nabla \times \tau$.

**Figure by using data**

[Figure]

Figure 2. Time series of Upwelling index (UI) and upwelling characteristics. (a) Time series of mean sea surface wind UI and wind stress curl in HEC region. Blue dotted curve denotes the mean UI during June-August; the red dotted curve is mean wind stress curl during June-August; and blue and red curves are the trends of the UI and wind stress curl, respectively. (b) Time series of upwelling area and SST. Green bar denotes the area of UEH region. Red and magenta dotted curve denote mean SST of UEH region and slope region (depth>200 m) in HEC, respectively. Blue, red and magenta curves are the trends of the upwelling area, mean SST in UEH and slope area, respectively.

Reference

Hersbach, H., Bell, B., Berrisford, P., Biavati, G., Horányi, A., Muñoz Sabater, J., Nicolas, J., Peubey, C., Radu, R., Rozum, I., Schepers, D., Simmons, A., Soci, C., Dee, D., Thépaut, J-N. (2018): ERA5 hourly data on single levels from 1959 to present. Copernicus Climate Change Service (C3S) Climate Data Store (CDS). (Accessed on < 15-Dec-2022 >), 10.24381/cds.adbb2d47.

Garratt, J.R., 1977. Review of Drag Coefficients over Oceans and Continents. Monthly Weather Review, 105, 915–929.

**[Major Comment 3]** Linear trends of wind upwelling index (UI), wind stress curl, SST and upwelling areas were estimated without assessing statistical significance of the trends. The trends

of SST, wind stress curl and upwelling areas might be insignificant in statistics because the r coefficients are small. Also note that the decreasing trend of UI from 2003-2020 was not presented in Fig. 2a.

**Response:** Thanks for your useful comment.

(1) We have added the *p* value into Figure 2 to indicate statistical significance of the trends. Because the period of data is only 18 years, it is a little too short to demonstrate the significance of the trend. Therefore, *p* value and *r* are not so statistically significant owing to the limitation of data. However, the trend is not the main result for this manuscript. We have mentioned this point in the manuscript.

(2) This time we use sea surface wind data from ERA5, which shows an increasing trend, from 0.45 to 0.55 $m^2 s^{-1}$. The wind stress curl shows an increasing trend, too. The information extracted from wind data indicates that the upwelling is enhancing during 2003-2020. However, the mean SST of UEH (core of upwelling) increased gradually from 2003 to 2020 as shown in Figure 2b. The increasing SST indicates that upwelling is weaking.

It seems that the trends of wind and SST are contradictory. We checked the mean SST of background (>200 m in HEC, magenta curve in Figure 2b). It shows that the SST of background increases much faster than that in UEH. Therefore, we conclude that the upwelling is enhanced by the stronger wind stress and curl, even though the SST of background increases much faster. We have added the trend of SST for background into Figure 2, and update the wind data in Figure 2a.

[Figure]

Figure 2. Time series of Upwelling index (UI) and upwelling characteristics. (a) Time series of mean sea surface wind UI and wind stress curl in HEC region. Blue dotted curve denotes the mean UI during June-August; the red dotted curve is mean wind stress curl during June-August; and blue and red curves are the trends of the UI and wind stress curl, respectively. (b) Time series of upwelling area and SST. Green bar denotes the area of UEH region. Red and magenta dotted curve denote mean SST of UEH region and slope region (depth>200 m) in HEC, respectively. Blue, red and magenta curves are the trends of the upwelling area, mean SST in UEH and slope area, respectively.

**[Major Comment 4]** Line 223-230: "Comparing the time series of Chl-a concentration shown in Figure 3 to the time series of upwelling characteristics shown in Figure 2, one can see that low UI values coincide with high Chl-a concentration in the UEH, and vice versa…". For the sake of visual comparison, the time series of Chl-a concentration and UI should be presented in the same figure. On the other hand, to gain a convincing result the out-of-phase relationship between the two-time series should also be quantified.

**Response:** Thanks for your useful comment. We have added the UI curve into Figure 3. As we can see, the peaks of Chl-a in summer (red curve) corresponded to valley of UI, especially before 2014. In 2018, there were minima for both of UI and Chl-a. However, the maximum of wind stress curl existed in 2018, which means the strongest wind stress curl was the leading factor for the upwelling process. The strongest wind stress curl generated a strong upwelling, and the Chl-a concentration is minimum in that year.

The relationship between UI, wind stress curl and Chl-a were very different in 2015, an unusual ENSO event. It looks like that the strong eastern-Pacific (EP) type caused this unusual high Chl-a concentration. Jing et al. (2011) had pointed out the analogous high Chl-a concentration in this study area in 1998. The correlation coefficient between wind UI and Chl-a concentration during 2003-2012 is -0.3, which shows a negative relationship. Because the limitation of data, the coefficient is a little small.

[Figure]

Figure 3. Time series of (a) the spatial mean of the Chl-a concentration in the upwelling area, (b) the monthly climatological mean, and (c) the seasonal mean of Chl-a and wind UI.

Reference:

Jing, Z., Qi, Y., and Du, Y., 2011. Upwelling in the continental shelf of northern South China Sea associated with 1997–1998 El Niño, Journal of Geophysical Research, 116, C02033.

**[Major Comment 5]** Line 470-490. "In the upwelling season, i.e., summer, the wind stress was smaller during El Nino events (Table 2) than during La Nina events (Figure 2a)...". I am confused. Previous studies (e.g. Wang et al (2001), Fang et al (2006), Huynh et al (2020)) showed that an

anomalous anticyclone (cyclone) develops over the western North Pacific during El Nino (La Nina) years and the summer southwesterly surface wind in the northwestern South China Sea can be enhanced after the El Nino peak. In Fig 2a, one can observe that the UI increased in 2005, 2010, and 2019, which correspond with the 2004-2005, 2009-2010 and 2018-2019 El Nino events. Therefore, the statement that the wind stress was smaller in the upwelling season during El Nino events might be incorrect. Did the authors discuss the relationships between Chl-a/wind stress and ENSO during the developing phase of ENSO, i.e. before ENSO peaks?

**Response:** Thanks very much for your comment.

(1) Yes, an anomalous anticyclone (cyclone) develops over the western North Pacific during El Nino (La Nina) years. However, the temporal period of anticyclone or cyclone is much shorter than one month. I think they are in different scales. Therefore, it is not paradoxical.

(2) The relationship between sea surface wind and El Nino in the northwestern South China Sea is a little complicate. Firstly, Hong and Zhang (2021) showed that the annual mean wind in different area in the northwestern South China Sea is different (Figure 5 with red legend) by using sea surface wind data of ERA5. The speed of sea surface wind was almost flat in station C (near HEC in this study) during 1981-1992. However, peaks and valleys could be seen clearly in Stations A, B and D during 1981-1992. Secondly, Yu et al. (2020) showed that the interannual variability indicates low levels of Chl-a southeast of Vietnam during El Niño years because of the weakened southwest monsoon. Huynh et al. (2020) also found that the western North Pacific anticyclone (cyclone) anomalously develops (Figures. 10b, e, c, f and 18b–d), leading to a weaker (stronger)-than-normal surface wind in the SCS in El Nino (La Nina) years. **These previous studies conclude that the weaking sea surface wind appears in El Nino years.**

There are some exceptions, i.e., 1997-1998 and 2015-2016. Fang et al. (2006) showed that the first time coefficient functions (TCFs) lags the Nino3.4 SST by 3 months during 1997-1998 event. In this study, the wind stress and curl were both strong during 2015-2016.

Moreover, La Nina event appears after El Nino event. The large time lag between Nino Index and sea surface wind could also confuse us. Therefore, I did not calculate the time lag in this study.

I think it depends on the type of El Nino. Maybe the eastern-Pacific (EP) type causes the strong wind during El Nino events in this study area. I have added the discussion into the manuscript. And, the helpful reference has added into the manuscript.

(3) I have added a figure to show the relationships between Chl-a, along-shelf wind and Nino Index.

[Figure]

Figure. Time series of Chl-a (green curve) and along-shelf wind (blue curve). Stripes point out the El Nino (magenta) and La Nina (blue) events. Black arrows point out the minima value of Chl-a concentration. Magenta dashed line indicate the high Chl-a concentration during El Nino events.

[Figure]

Figure 5. Trends of annual mean wind speed (a) and strong wind days (b) at station A, B, C and D, respectively, from 1979 to 2019. The annual mean wind speed and corresponding trend obtained from ERA5 data at station A were compared with the data obtained from the adjacent Macau Airport (from 1997 to 2015). Stations A and B are in the Pearl River Estuary. Station C is in Yuexi (near

HEC in this study). Station D is in Yuedong. *Cited from Hong and Zhang (2021).*

References

Hong B, Zhang J., 2021. Long-Term Trends of Sea Surface Wind in the Northern South China Sea under the Background of Climate Change. Journal of Marine Science and Engineering. 9(7):752. https://doi.org/10.3390/jmse9070752

Huynh, HN.T., Alvera-Azcárate, A. Beckers, JM. Analysis of surface chlorophyll a associated with sea surface temperature and surface wind in the South China Sea. Ocean Dynamics 70, 139–161 (2020). https://doi.org/10.1007/s10236-019-01308-9

Yu, Y., Y. Wang, L. Cao, F. Chai, 2020. The ocean-atmosphere interaction over a summer upwelling system in the South China Sea. Journal of Marine Systems, 208, 103360.

**[Major Comment 6]** Line 496-499."The Chl-a concentration in summer was mainly regulated by upwelling processes (Jing et al., 2011), with a negative correlation (Figures 2–3). Therefore, the increased precipitation and weaker upwelling processes could have induced the increased Chl-a concentration in the HEC (upward arrow in Figure 12)." Why does the summer upwelling have a negative correlation with the Chl-a concentration in the HEC?

**Response:** The upwelling in HEC is control by alongshore wind and wind stress curl (Hu and Wang, 2016). Strong (weak) southwesterly wind and positive curl would induce strong (weak) upwelling. From Figure 2-3, one can see that minimum (2005, 2008, 2012, 2015 and 2018) Chl-a concentration appears with maximum wind UI or wind stress curl (strong wind stress curl in 2018) during upwelling season (red curve for Chl-a and magenta curve for wind UI). The strong (weak) upwelling exists with low (high) Chl-a concentration. Therefore, the summer upwelling has a negative correlation with the Chl-a concentration in the HEC.

[Figure]

Figure 3. Time series of (a) the spatial mean of the Chl-a concentration in the upwelling area, (b) the monthly climatological mean, and (c) the seasonal mean of Chl-a and UI.

Reference:

Hu JY, XH Wang, 2016. Progress on upwelling studies in the China seas. Reviews of Geophysics, 54(3):653-673.

**[Major Comment 7]** Line 533-536. "There was a positive correlation between the Chl-a anomalies and the La Nina events…". The quantitative correlation between Chl-a and ENSO should be estimated.

**Response:** Thanks very much for your comment. Because La Nina event appears after El Nino event, therefore, a large time lag between Nino Index and sea surface wind or Chl-a concentration could confuse their relationships. I have added figure to show the relationships between Chl-a and along-shelf wind.

[Figure]

Figure. Time series of Chl-a (green curve) and along-shelf wind (blue curve). Stripes point out the El Nino (magenta) and La Nina (blue) events. Black arrows point out the minima value of Chl-a concentration. Magenta dashed line indicate the high Chl-a concentration during El Nino events.

Moreover, as an attached drawing for this response, cross wavelet analysis is used to indicate the relationship between wind and Chl-a concentration. The main period band is 1 year. For most of time, Chl-a lags wind about 3 months. During 2015-2016, the time lag is about 6 months.

[Figure]

Attached Figure. Cross wavelet analysis between along-shelf wind and Chl-a concentration. The thick line is the 5% significance level against red noise and the core of influence is shown as the thin line. The arrows show the relative phase relationship between two time series with in-phase (anti-phase, leading and laging) pointing right (left, down and up).

**[Minor Comment 1]** Line 182-189. The caption does not totally correspond to Fig. 2.
**Response:** Thanks very much for your comment. I have revised the caption as follows.

"Figure 2. Time series of Upwelling index (UI) and upwelling characteristics. (a) Time series of mean sea surface wind UI and wind stress curl in HEC region. Blue dotted curve denotes the mean UI during June-August; the red dotted curve is mean wind stress curl during June-August; and blue and red curves are the trends of the UI and wind stress curl, respectively. (b) Time series of upwelling area and SST. Green bar denotes the area of UEH region. Red and magenta dotted curve denote mean SST of UEH region and slope region (depth>200 m) in HEC, respectively. Blue, red and magenta curves are the trends of the upwelling area, mean SST in UEH and slope area, respectively."

[Minor Comment 2] Line 215. Change "climatologic" to "climatological"

Response: Thanks very much for your comment. I have changed the word "climatologic" into "climatological".

[Minor Comment 3] Line 244-246. "... (d) euphotic depth and TSS in the study area…" should be "...(d)euphotic depth, TSS and Chl-a concentration in the study area…"

Response: Thanks very much for your comment. I missed the word "Chl-a". I have revised the caption as the suggestion.

[Minor Comment 4] Line 227-228. "Moreover, one can see that the Chl-a concentration is unexpectedly low in the upwelling season, as shown in Figures 2a-b" should be "... in Fig. 3"

Response: Thanks very much for your comment. I have corrected it.

[Minor Comment 5] Line 484-485. An El Nino (La Nina) often takes place between two calendar years. However, most of the El Ninos listed in Table 1 are single year events, for example the 2015 El Nino which actually occurred from October 2014 to April 2016 (https://origin.cpc.ncep.noaa.gov/products/analysis_monitoring/ensostuff/ONI_v5.php). Please correct the data in Table 1.

Response: Thanks very much for your comment. I have revised the Table 1. I have added figure to show the relationships between Chl-a and along-shelf wind, which is more clearly than Table 1.

[Figure]

Figure. Time series of Chl-a (green curve) and along-shelf wind (blue curve). Stripes point out the El Nino (magenta) and La Nina (blue) events. Black arrows point out the minima value of Chl-a concentration. Magenta dashed line indicate the high Chl-a concentration during El Nino events.

---

## Referee Report (RR1)

**Comment on "Multiple mechanisms for chlorophyll-a concentration variations in coastal upwelling regions: A case study east of Hainan Island in the South China Sea" - the revised version, by Junyi Li et al.**

The manuscript is improved. However, the authors should make some revisions before the manuscript can be published. My comments are listed below.

1. In section 3.2 the explanation about the insignificance of trends, from line 260 to 262, should appear after line 280 or at the end of the section, because not only the trends of UI and WSC but also the trends of upwelling area and UEH SST are not statistically significant. Besides, please change "HEU" in Fig. 4b to "UEH".

2. Line 312-313: -0.3 is the correlation coefficient between wind UI and Chl-a from 2003-2012 or 2003-2020? If for 2003-2012, how about the relationship between wind UI and Chl-a from 2013-2020?

3. Line 494-500: "During El Niño events, the weakened southwesterly monsoon suppresses ocean upwelling (Jing et al., 2011; Kuo et al., 2008). The reverse occurs during La Niña events. Jing et al. (2011) found that the significantly strengthened wind stress of the 1998 summer induced strong upwelling, and the Chl-a concentration was much higher than in any of the other years. Yu et al. (2020) showed that the interannual variability indicates low levels of Chl-a southeast of Vietnam during El Nino years because of the weakened southwest monsoon. These previous studies conclude that the weakening sea surface wind appears in El Nino years". Note that in Jing et al. (2011)' work, they demonstrated that the wind stress strengthened in the continental shelf of the northern South China Sea (SCS), particularly in the eastern Hainan Island - the same area of this study, in summer 1998 associated with the 1997-1998 El Nino, not La Nina. They also indicated that due to the anticyclonic atmospheric circulation anomaly over the SCS and northwest Pacific, the local southwesterly winds in the northwestern SCS are dramatically enhanced, whereas the southwesterly winds are weakened in the central and western SCS (Fig. 8 in their work). Jing et al., (2011) is contrary to the conclusion drawn in the last sentence as well as the author's response (1) to the fifth major comment.

4. Regarding the author's response (2) to the fifth major comment, note that Fang et al. (2006) and Huynh et al. (2020) also showed that the spatial variability of the monsoon winds under the El Nino (La Nina) effect is not in-phase in subregions of the SCS. In Hong and Zhang (2021), station C is located off the northeastern Leizhou Peninsula; Fig. 2b in their work shows that the trends of the annual mean wind speed between station C and the eastern Hainan are out-of-phase from 1979 to 2019, decreasing trends at station C, whereas increasing trends in eastern Hainan. A strong increasing trend of positive wind stress curl was detected in eastern Hainan during 1979-2019 (Fig. 10). Additionally, Hong and Zhang (2021) also indicated that the surface wind speed in eastern Hainan has a positive correlation with ENSO (Figs. 8-9).

---

## Author Response (AR2)

Dear Editor and Reviewers

We are very pleased to have your comments concerning our manuscript entitled "Multiple mechanisms for chlorophyll-a concentration variations in coastal upwelling regions: A case study east of Hainan Island in the South China Sea" (egusphere-2022-969). Thank the editor and reviewers for taking time out of your busy schedule to review our paper and provide constructive comments on it.

We have read and dealt with all the comments carefully. The revised manuscript with all comments highlighted with blue fronts has been uploaded, and point-to-point responses to the reviewer's comments are present following. Furthermore, reference list and color schemes used in figures (Figs 2d, 4b and 5c) have been compiled.

**Response to Comments of Reviewer 1** (Blue font in the manuscript)**

[Comment 1] In section 3.2 the explanation about the insignificance of trends, from line 260 to 262, should appear after line 280 or at the end of the section, because not only the trends of UI and WSC but also the trends of upwelling area and UEH SST are not statistically significant. Besides, please change "HEU" in Fig. 4b to "UEH".

**Response:** Thank you for your careful reading.

(1) We have removed the sentence to the end of this section.

**Lines 276-278**

"Because the period of data is only 18 years, it is a little too short to demonstrate the significance of the trend of wind and SST data. Therefore, p value and r are not so statistically significant owing to the limitation of data."

**(2) We have corrected the word "HEU" in Fig. 4b.**

Figure 4. Time series of Upwelling index (UI) and upwelling characteristics. (a) Time series of mean sea surface wind UI and wind stress curl in HEC region. Blue dotted curve denotes the mean UI during June-August; the red dotted curve is mean wind stress curl during June-August; and blue and red curves are the trends of the UI and wind stress curl, respectively. (b) Time series of upwelling

area and SST. Green bar denotes the area of UEH region. Red and magenta dotted curve denote mean SST of UEH region and slope region (depth>200 m) in HEC, respectively. Black, red and magenta curves are the trends of the upwelling area, mean SST in UEH and slope area, respectively.

[Comment 2] Line 312-313: -0.3 is the correlation coefficient between wind UI and Chl-a from 2003-2012 or 2003-2020? If for 2003-2012, how about the relationship between wind UI and Chl-a from 2013-2020?

**Response:** Thank you for your careful reading.

(1) A negative value, -0.3, is the correlation coefficient for the data from 2003 to 2012.

(2) The correlation coefficient during 2013-2020 is about 0.2. Firstly, as we can see in Fig.4, the wind stress curl is more important than wind stress for upwelling in 2018. Secondly, a typical strong ENSO event occurred in 2015-2016. Therefore, the relationship between wind UI and Chl-a from 2013 to 2020 looks like irrelevant. We have added a sentence to improve the description. Lines 307-308.

"The main reason for the negative relationship is the low background SST during 2003-2012." Lines 314-318.

"The relationship between wind UI and Chl-a concentration looks like irrelevant during 2013-2020. The main reasons are the strong ENSO event in 2015-2016 and the strong wind stress curl in 2018. High wind UI and wind stress curl occurred in 2015-2016 combined with high Chl-a concentration. In 2018, a strong wind stress curl with weak wind UI induced a strong upwelling process (low Chl-a concentration) as shown in Figure 4."

**[Comment 3]** Line 494-500: "During El Niño events, the weakened southwesterly monsoon suppresses ocean upwelling (Jing et al., 2011; Kuo et al., 2008). The reverse occurs during La Niña events. Jing et al. (2011) found that the significantly strengthened wind stress of the 1998 summer induced strong upwelling, and the Chl-a concentration was much higher than in any of the other years. Yu et al. (2020) showed that the interannual variability indicates low levels of Chl-a southeast of Vietnam during El Nino years because of the weakened southwest monsoon. These previous studies conclude that the weakening sea surface wind appears in El Nino years". Note that in Jing et al. (2011)' work, they demonstrated that the wind stress strengthened in the continental shelf of the northern South China Sea (SCS), particularly in the eastern Hainan Island - the same area of this study, in summer 1998 associated with the 1997-1998 El Nino, not La Nina. They also indicated that due to the anticyclonic atmospheric circulation anomaly over the SCS and northwest Pacific, the local southwesterly winds in the northwestern SCS (Fig. 8 in their work). Jing et al., (2011) is contrary to the conclusion drawn in the last sentence as well as the author's response (1) to the fifth major comment.

**Response:** We gratefully appreciate for your valuable suggestion. We have checked the references carefully, and found the El Nino would enhance the wind stress in the eastern Hainan area. Moreover, there is time lag between El Nino and wind stress, about several months. In weak El Nino event, e.g., 2007 and 2010, the duration of El Nino is a little short, and it seems that the strong wind occurs in the consecutive La Nina event. In the typical El Nino event, i.e., 2015, it began in the winter of 2014, and the strong wind occurred in the summer of 2015. The strong wind induced a strong upwelling. Therefore, the Chl-a concentration was low in summer of 2015. We have corrected the

**manuscript.**

Lines 491-494

"During El Niño events, the positive southwesterly wind anomalies would enhance the coastal upwelling in the SCS (Jing et al., 2011; Kuo et al., 2008). The positive southwesterly wind anomalies lag El Niño event several months (Hong and Zhang, 2021; Huynh et al., 2020). The reverse occurs during La Niña events."

**Lines 496-503**

"In 2005, the wind stress and upwelling area were much larger than that in 2004. And the Chl-a concentration decreased to 0.6 mg m–3 in June 2005. During 2015-2016, the wind stress and curl were both strong and the upwelling area was larger than that in 2014. There was anomalously high Chl-a concentration occurred in 2016. Jing et al. (2011) have reported analogously high Chl-a concentration anomaly in 1998. We should notice that a maximum SST occurred in 2016 with a maximum of Chl-a concentration. While, there were minimum value of background SST (Figure 4b) occurred in summer of the year 2008, 2011, 2012, 2017 and 2018 combined with minimum of Chl-a concentration (arrows in Figure 12)."

**[Comment 4]** Regarding the author's response (2) to the fifth major comment, note that Fang et al. (2006) and Huynh et al. (2020) also showed that the spatial variability of the monsoon winds under the El Nino (La Nina) effect is not in-phase in subregions of the SCS. In Hong and Zhang (2021), station C is located off the northeastern Leizhou Peninsula; Fig. 2b in their work shows that the trends of the annual mean wind speed between station C and the eastern Hainan are out-of-phase from 1979 to 2019, decreasing trends at station C, whereas increasing trends in eastern Hainan. A strong increasing trend of positive wind stress curl was detected in eastern Hainan during 1979-2019 (Fig. 10). Additionally, Hong and Zhang (2021) also indicated that the surface wind speed in eastern Hainan has a positive correlation with ENSO (Figs. 8-9).

**Response: Thanks for your useful comment.**

Yes, we agree with spatial variability of the monsoon winds under the El Nino (La Nina) effect. Figs. 2b and 10 in Hong and Zhang (2021) showed an increasing trend of wind speed and wind stress in the eastern Hainan area, which is the same with the Fig. 4a in this study.

Fig. 8 in Hong and Zhang (2021) showed a positive wind speed anomaly for Mode 1 and 3, while a negative anomaly in the eastern Hainan area in summer of 1997. In the summer of 1998, the positive wind speed anomaly occurred in all the three Modes in the eastern Hainan area. The figure showed a negative wind speed anomaly in the summer of 1999 and 2000. In 2015, the positive wind speed anomaly occurred once again. Moreover, from Fig. 9, one can see that the wind lags Nino index several months. Therefore, the surface wind speed in eastern Hainan has a positive correlation with El Nino.

We have corrected the manuscript about the relationship between ENSO and wind UI. The higher (lower) wind speed occurs during El Nino (La Nina) event in the eastern Hainan Island. And, we have added the reference into the manuscript.

**Lines 491-494**

"During El Niño events, the positive southwesterly wind anomalies would enhance the coastal upwelling in the SCS (Jing et al., 2011; Kuo et al., 2008). The positive southwesterly wind anomalies lag El Niño event several months (Hong and Zhang, 2021; Huynh et al., 2020). The reverse occurs during La Niña events."

---

## Author Response (AR3)

Dear Editor

Many thanks for your time. We have revised 8 figures in the manuscript (Figures 1, 2, 4, 5, 10, 11, 12 and 13) according to the simulator. Markers and legends have been also added into the figures, which is more convenient to follow. Moreover, authors have polished the manuscript marked by the blue font.

Looking forward to hearing from you. Have a good weekend!

Best regards,
Junyi Li
March 10, 2023

**Details of modification**

**(1) The revised manuscript:**

Lines 314-320:

[revised manuscript text omitted]